# Whole-Genome Resequencing Highlights the Unique Characteristics of Kecai Yaks

**DOI:** 10.3390/ani12192682

**Published:** 2022-10-06

**Authors:** Yandong Kang, Shaoke Guo, Xingdong Wang, Mengli Cao, Jie Pei, Ruiwu Li, Pengjia Bao, Jiefeng Wang, Jiebu Lamao, Dangzhi Gongbao, Ji Lamao, Chunnian Liang, Ping Yan, Xian Guo

**Affiliations:** 1Key Laboratory of Yak Breeding Engineering of Gansu Province, Lanzhou Institute of Husbandry and Pharmaceutical Sciences, Chinese Academy of Agricultural Sciences, Lanzhou 730050, China; 2Key Laboratory of Animal Genetics and Breeding on Tibetan Plateau, Ministry of Agriculture and Rural Affairs, Lanzhou 730050, China; 3Station of Animal Husbandry in Xiahe County, Xiahe 747199, China

**Keywords:** Kecai yak, whole-genome resequencing, genetic diversity, molecular phylogeny

## Abstract

**Simple Summary:**

Kecai yaks are an important genetic resource in Gansu Province, China. These yaks, produced through the crossbreeding of Gannan yaks and wild yaks, exhibit excellent production performance and high fecundity. Despite these beneficial characteristics, the genetic characteristics, population structure, and other properties of Kecai yaks have not been effectively characterized to date. Accordingly, this study was developed to explore the genetic diversity and population structure of Kecai yaks through whole-genome resequencing. These analyses ultimately revealed the status of Kecai yaks as an independent group within the overall Chinese yak population.

**Abstract:**

Kecai yaks are regarded as an important genetic resource in China owing to their high fecundity and flavorful meat. However, the genetic characteristics of Kecai yaks have not been effectively characterized to date, and the relationship between Kecai yaks and other yak breeds remains to be fully characterized. In this paper, the resequencing of the Kecai yak genome is performed leading to the identification of 11,491,383 high-quality single nucleotide polymorphisms (SNPs). Through principal component, phylogenetic, and population genetic structure analyses based on these SNPs, Kecai yaks were confirmed to represent an independent population of yaks within China. In this study, marker and functional enrichment analysis of genes related to positive selection in Kecai yak was carried out, and the results show that such selection in Kecai yaks is associated with the adaptation to alpine environments and the deposition of muscle fat. Overall, these results offer a theoretical foundation for the future utilization of Kecai yak genetic resources.

## 1. Introduction

Yaks (*Bos grunniens*) are a long-haired livestock species native to the Qinghai–Tibet Plateau and surrounding regions, living at an average altitude of over 3000 m. These yaks are uniquely adapted to high-altitude conditions such that they exhibit robust cardiopulmonary functionality, strong muscles, and good feeding performance [1]. Yaks are a critical source of animal labor, food, and other materials for local herdsmen such that they are an integral facet of life on the Qinghai–Tibet Plateau. Kecai yaks were domesticated from wild yak (*Bos mutus*) populations, and have accordingly experienced a series of population bottlenecks, artificial selection events, and continuous bidirectional gene flow [2,3].

Several studies have explored yak population structures and performed detailed analyses of the yak domestication process based on mtDNA sequences and other nucleic acid markers. By analyzing mitochondrial D-loop sequences from 250 domesticated yaks and 13 wild yaks, Guo et al. [3] were able to identify two distinct maternal branches that diverged approximately 130,000–150,000 years ago. Through further comparisons of mitochondrial sequences from 405 domesticated yaks and 45 wild yaks, Wang et al. [4] defined a third maternal branch that only included wild yaks and further observed the effects of relaxed selection among domesticated yaks as a consequence of domestication. Wang et al. [5] further compared mitochondrial sequence data from five populations of yaks in Xinjiang and Qinghai, ultimately separating these domestic and wild yaks into three clades and proposing an update to the yak reference genome to support further research efforts. In 2012, Qiu et al. [6] sequenced and assembled the first domestic yak genome through the use of second-generation DNA sequencing technologies, gaining preliminary insight into the potential genetic basis for the ability of yaks to adapt to high-altitude living. More recently, Liu et al. [7] employed a second-generation sequencing approach to compile the first wild yak genome, thereby offering a valuable resource for further molecular biology-focused studies. Ji et al. [8] additionally leveraged PacBio long-read sequencing technologies to compile the first chromosome-level domesticated yak reference genome, enabling the in-depth analysis of genetic diversity among different yak breeds. Through the use of resequence samples from 65 domesticated yaks and 14 wild yaks, Zhang et al. [9] detected 2643 copy number variation regions (CNVRs), ultimately leading to the identification of relationships between domestication traits and variations in certain protein-coding genes, including the brain-development-related *GRIN2D* gene. Zhang et al. [10] were further able to identify genes associated with immunity, behavior, reproduction, and the nervous system through comparisons of structural variations (SVs) between domesticated and wild yaks.

Gannan yaks are 1 of 18 unique genetic resources in China, exhibiting advantageous characteristics including robust adaptability, nutrient-rich meat, and a palatable taste [11]. Kecai yaks were produced through several generations of crossing wild yaks with Gannan yaks. These Kecai yaks are primarily bred in Kecai Town, Xiahe County, Gannan Autonomous Prefecture, Gansu Province, and are characterized by a black body with horns and fluffy hair located on the chest and under the tail (Figure 1a). These Kecai yaks are large, exhibit high fecundity, and inherited the excellent characteristics of delicious meat of the Gannan yak. Despite their advantageous characteristics and status as an important domestic yak genetic resource, the genetic characteristics, unique traits, and population structure of Kecai yaks have not been studied. Accordingly, this study was developed with the goal of exploring Kecai yak population structure and genetic diversity through genomic resequencing and the comparison of these genomic sequences with those from other yak breeds. The overall goal of this approach is to provide a theoretical foundation for the future use of Kecai yak genetic resources based on a genome-level understanding of these animals.

## 2. Materials and Methods

### 2.1. Ethics Statement

The China Council on Animal Care and the Ministry of Agriculture of the People’s Republic of China guidelines were followed when performing all animal studies. The Animal Care and Use Committee of the Lanzhou Institute of Husbandry and Pharmaceutical Sciences Chinese Academy of Agricultural Sciences approved all yak handling protocols (Permit No: SYXK-2014-0002). Blood samples for this study were obtained from the jugular vein of live yaks following surface disinfection with topical alcohol. Iodophor was applied to the site of venipuncture after blood collection was complete to prevent infection.

### 2.2. Animals Sample Collection

For this study, 45 adult yaks of similar age, health status, and size were analyzed. A total of 9 Kecai yaks (Kecai Town, Gannan Tibetan Autonomous Prefecture, Gansu Province, China), 7 Gannan yaks (Xiahe County, Gannan Tibetan Autonomous Prefecture, Gansu Province, China), 10 Jiulong yaks (Jiulong County, Ganzi Tibetan Autonomous Prefecture, Sichuan Province, China), 10 Tianzhu white yaks (Tianzhu Tibetan Autonomous Prefecture, Gansu Province, China), and 9 Jinchuan yaks (Aba Tibetan and Qiang Autonomous Prefecture, Sichuan Province, China) were included. DNA was extracted from individual blood samples with an EasyPure Blood Genomic DNA Kit (Quanshijin Biotechnology Co., Ltd., Beijing, China). DNA purity and concentrations were measured with a Nanodrop 2000 spectrophotometer (Thermofisher Scientific, MA, USA) based on OD260/280 values. The integrity of samples with an OD260/280 of 1.6–1.8 and a concentration of 50–1000 ng/µL were separated via 1% agarose gel electrophoresis (AGE) to ensure sample quality. Those samples that exhibited a single distinct AGE band were diluted to 50 ng/µL and stored at −80 °C. For sequencing analyses, these samples were fragmented into 350–500 bp segments with Covaris after which a TruSeq DNA LT Sample Prep kit was used for library construction. Briefly, DNA fragments were subjected to end repair, polyadenylation, the addition of sequencing connectors, purification, PCR amplification, and other necessary steps, with the final libraries then being used for 150 bp paired-end sequencing with the HiSeq X Ten platform (Illumina Inc., Hayward, CA, USA). Wild yak genomic data were obtained from breeding genome data generated by the European Nucleotide Archive (EMBL-EBI) (Accession number: PRJNA285834), while Kecai yak sequencing data were uploaded to the Sequence Read Archive (SRA) (Accession number: PRJNA842787). The data were compared to genomic data from other local yak populations.

### 2.3. Sequence Quality Control and Filtering

Raw reads from Kecai yaks were filtered to remove low-quality reads, base duplications, and other forms of artificial bias. Library sequencing and data analyses were partially conducted by Shanghai OE Biomedical Technology Co. Raw reads were initially obtained in the fastq format following high-throughput sequencing, after which pre-processing was performed using fastp v 0.20.0 (Shenzhen, China) [12] through four primary steps: (1) the removal of spliced sequences; (2) the removal of reads with ≥ 5 N (non-AGCT) bases; (3) the removal of reads with an average base mass value < 20 over a sliding window of 4 bases; and (4) the removal of remaining reads < 75 bp in length or with an average base mass value < 15.

### 2.4. Genome Alignment

Reference genome alignment of the filtered reads was performed using BWA v 0.7.12 [13], using the BWA mem alignment algorithm with the default parameters. The results were formatted and sorted using SAMtools v 1.9 [14], after which duplicate reads were identified using the MarkDuplicates function Picard v 2.18.17 (https://broadinstitute.github.io/picard/) (accessed on 15 January 2022). To enable statistical analyses, the results were then compared with Qualimap [15].

### 2.5. Variant Detection and Annotation

SNP and InDel detection were performed using the Haplotypecaller module in GATK v.3.8.1 [16] based on comparisons between sample sequences and the reference genome (LU_Bosgru_v3.0). For each sample, gvcf files were generated and SNPs were detected using the GenotypeGVCFs module. Data were filtered with the following criteria: (1) The Genome Analysis Toolkit (GATK) filtering parameters for loci exclusion: QualByDepth (QD) < 2.0 || RMSMappingQuality (MQ) < 40.0 || FisherStrand (FS) > 60.0 || StrandOddsRatio (SOR) > 3.0 || MappingQualityRankSumTest (MQRankSum) < −12.5 || ReadPosRankSum < −8.0; (2) Allele type: as SNPs are generally biallelic, loci with more than 2 distinct genotypes were filtered out; (3) Loci with a minor allele frequency (MAF) < 0.05 were removed.

GATK was similarly used to analyze InDels in these data with the following screening criteria: QD < 2.0 || MQ < 40.0 || FS > 200.0 || SOR > 3.0 || MQRankSum < −12.5 || ReadPosRankSum < −8.0.

The SnpEff program was used to annotate InDels and SNPs based on the reference genome [17]. Annotation was only performed for high-quality SNPs. SNPs were classified as being located in exonic, intronic, intergenic, upstream, downstream, or splicing regions of the genome. Those SNPs located within coding exons were additionally classified as either synonymous or non-synonymous. Variants resulting in stop-gain or stop-loss mutations were additionally annotated. InDels were classified based on whether they were conserved in-frame insertions/deletions, disruptive in-frame insertions/deletions, bidirectional gene fusions, frameshift mutations, or conservative–disruptive in-frame insertions/deletions. SNPs were specifically annotated with classifications including missense, initiator codon, and synonymous variants.

### 2.6. Population Genetic Polymorphism Analyses

To facilitate a genome-wide analysis of phylogenetic relationships, PLINK [18] was used to identify SNPs in linkage disequilibrium (LD) using the following parameters: ‘--indep-pairwise 50 10 0.2’. Those SNPs not exhibiting close linkage were selected, after which neighbor-joining (NJ) phylogenetic trees were generated with the NEIGHBOR program in PHYLIP (http://evolution.genetics.washington.edu/phylip.html) (accessed on 15 January 2022). Feature vectors were computed with EIGENSOFT [19], which was used to extract the first four principal components for a principal component analysis (PCA). Population structure analyses were performed using ADMIXTURE [20] across a range of k values from 2 to 10, with 10 different seeds being used for 10 repeated analyses. The results were clustered using Pong [21], and optimal k values were determined based upon cross-validation error. The PopLDdecay program [22] was employed to calculate LD with the following settings: ‘-MaxDist 500 -MAF 0.05’.

### 2.7. Selective Sweep and Functional Enrichment Analyses

Selected regions of the genome were identified by using the θπ ratio and Fst values for screening purposes [23], as this strategy can effectively identify sites under selective pressure, particularly when mining for functional regions related to environmental stress that often yield strong selection signals. The log_2_θπ ratio and Fst values were calculated using VCFtools [24], with a window size of 100 kb and a step size of 10 kb. Those regions in the top and bottom 5% of log_2_θπ ratio values and the top 5% of Fst values were identified as selected regions of the genome, and the target genes were subject to functional enrichment analyses using Gene Ontology (GO) and Kyoto Encyclopedia of Genes and Genomes (KEGG) functional annotation tools. The numbers of genes associated with these GO terms and KEGG pathways were counted, after which a hypergeometric distribution test was used to compute the significance of gene enrichment therein.

## 3. Results

### 3.1. Sequencing, Detection, and Annotation of Genome-Wide SNPs and InDels

In these resequencing analyses, an average of 209,142,764 raw reads were obtained per Kecai yak sample, with 208,238,267 valid reads (≥99%) being retained following the filtering of unqualified data. Overall, 95.23% of reads with a quality value ≥20 bases (sequencing error rate < 0.01), with a GC content of 43.14%, with good sequencing quality. After filtering, over 98.75% of these reads were successfully aligned to the reference genome (Appendix A).

The genomes of 13 wild yaks and 45 domestic yaks from the Gansu and Sichuan provinces were analyzed. After filtering, 11,491,383 valid high-quality SNPs were retained for subsequent annotation and analysis. Resequencing was achieved at an average 6.7× depth with 99.07% average genome coverage. Intergenic regions accounted for the majority (64.77%) of these data (Appendix A), while intronic regions accounted for 32.18% of the data. Exonic regions accounted for just 0.81% of the overall dataset (Figure 2a), including 46,970 and 45,006 synonymous and nonsynonymous mutations, respectively (Appendix A). The ratio of transition mutations (n = 31,935,918) to transversion mutations (n = 12,776,170) was 2.499, with T > C and A > G being the most common substitution mutation types (Figure 2d).

Following the filtering of InDel data for all yak populations, 1,404,453 InDel loci were obtained and associated genes were annotated. Of these InDels, 63.35% of the indels are located in the intergenic region, 33.13% in the intron region, and only 0.30% in the exon region (Figure 2b). Among the indels located in the exon region, 1217 induced frameshift deletions, 2108 induced frameshift insertions, and 432 and 342, respectively, induced non-frameshift deletions and non-frameshift insertions (Appendix A). Deletions and insertions of less than 4 bp accounted for 82.87% and 91.28%, respectively (Figure 2e). SNP and InDel distributions across Kecai yak chromosomes are represented with Circos plots shown in Figure 2c.

### 3.2. Population-Level Analyses of Genetic Polymorphisms and LD Decay

To explore the genetic population structure of Kecai yaks and their relationship with other yak populations, genome-wide polymorphism analyses were performed for both domesticated and wild yaks. Phylogenetic trees developed based upon these genome-wide resequencing data indicated that each group of yaks clustered separately, with Kecai yaks being located immediately between wild and Gannan yaks, with some Kecai yaks being genetically similar to wild yaks but wholly separate from Gannan yaks, consistent with their identity as a separate group (Figure 3e). In a principal component analysis, samples from Kecai yaks, Tianzhu white yaks, Jiulong yaks, and wild yaks were grouped together (Figure 3a). Specifically, Kecai yak samples clustered in close proximity to those from Jinchuan and Gannan yaks, suggesting that they share a similar genetic background (Figure 3a). In further analyses of population structure, at k = 2, all of yaks were separated into two groups: wild yaks and other yaks. When k = 3, Jiulong yaks were classified separately from other yaks. When k ≥ 4, Gannan yaks were similarly separated from other yaks, and Gannan and Kecai yaks were separated into distinct groups (Figure 3d). LD decay patterns in these different yak populations were analyzed based on the correlations between allele frequencies and genomic distances (r2) for SNP pairs. In this analysis, the slowest decay was observed for Gannan yaks, followed by Jiulong and Xueduo yaks, whereas the most rapid decay was observed for Kecai yaks. These results suggest that, relative to these other yak populations, Kecai yaks have been subject to a lesser degree of domestication and to less intense selection (Figure 3f).

### 3.3. Analyses of Genetic Diversity

The genetic differentiation index (Fst) is frequently used in population genetics studies as a means of gauging the degree of genetic differentiation between two populations. When comparing the Xueduo and Kecai yak populations, the average Fst value was 0.00523, whereas it rose to 0.01998 when comparing Gannan and Kecai yaks. Xueduo yaks are primarily distributed throughout the Henan County located in Huangnan Prefecture, Qinghai Province, in the eastern region of the Qinghai–Tibet Plateau. Ganna yaks are primarily distributed in the Gannan Tibetan Autonomous Prefecture regions of Gansu Province. Greater differentiation was evident between Gannan and Kecai yaks relative to that observed between Xueduo and Kecai yaks. Higher levels of nucleotide diversity correspond to a higher degree of genetic variation within a given population. In these resequencing analyses, nucleotide diversity was 0.00121 in Gannan yaks, 0.00117 in Jiulong yaks, 0.00132 in Kecai yaks, and 0.00127 in Xueduo yak, suggesting that Kecai yaks exhibit the highest levels of nucleotide diversity among analyzed yak populations.

The θπ ratio and Fst values were next used to screen the areas of the genome under selection using defined thresholds (log2θπ ratio > 0.668, Fst ≥ 0.086); values above these thresholds indicate positive selection and can be divided into groups with unique characteristics. When comparing Gannan and Kecai yak samples, the selection area under positive selection was found to be greater than that under negative selection (Figure 4b).

### 3.4. Enrichment Analysis

Genome-wide variations in the Gannan and Kecai yak populations were next analyzed, with the results being used for GO enrichment analyses of the 1882 identified genes (Appendix A). These genes were enriched in biological processes including cellular processes, metabolic processes, and responses to stimuli, cellular components including organelles, macromolecular complexes, synaptic fractions, and molecular functions, including catalytic activity, enzyme regulator activity, and molecular transducer activity (Figure 4a). In KEGG enrichment analyses, these genes were found to be closely associated with several biologically important pathways (Appendix A), including the Staphylococcus aureus infection, terpene skeleton biosynthesis, sphingolipid metabolism, linoleic acid metabolism, and FcγR-mediated phagocytosis pathways (Figure 4c). Additionally, in the pathways involved in muscle fat development, a candidate gene *ACOX1* related to fat deposition was found.

## 4. Discussion

Yaks are an invaluable genetic resource in the Qinghai–Tibet Plateau and surrounding high-altitude regions. Several reports estimate that yaks were initially domesticated from wild yak populations approximately 7300 years ago [25,26]. Gannan yaks represent a local genetic resource unique to the Qinghai–Tibet Plateau. Following prolonged natural selection and artificial breeding, these yaks exhibit a stable population structure, robust stress resistance, and excellent reproductive performance. Gannan-yak-derived products have a geographical indication and are highly desirable owing to their production of high-quality milk and meat [11]. Kecai yaks are a recently described genetic resource generated through the breeding of Gannan yaks with wild yaks. These Kecai yaks are larger than other yak breeds, and retained the excellent reproductive and production performance of Gannan yaks. To better understand the evolution and genetic characteristics of these Kecai yaks in the present study, genomic resequencing was performed and the resultant data were compared with genomic sequences from other yaks to establish the overall yak population structure. In addition, the genetic diversity of Kecai yaks was analyzed, and the unique characteristics of Kecai yaks and other yak breeds were evaluated.

A phylogenetic tree was constructed by SNP, and it was found that wild yak and domestic yak were mainly divided into two different branches. Because Kecai yak is the offspring of a branch of Gannan yak crossed with wild yak, some Kecai yak are genetically close to wild yak. The genetic similarities between wild and Kecai yak populations are likely attributable to the fact that Kecai yaks were genetically distinguishable from wild and Gannan yak populations. When analyzing yak population structures at k = 2, these yaks were clustered into wild and domesticated yak samples in line with prior research [25,27]. When k ≥ 4, Gannan yak samples formed an independent subgroup distinct from other domesticated yak samples. Compared with Gannan yak, Jiulong yak and Xueduo yak, the measured Fst values of harvestable yak were all lower than 0.05, indicating that there was gene exchange among yak populations. Notably, Kecai yak samples exhibited a higher degree of nucleic acid diversity relative to other domestic yak subpopulations. These characteristics offer insights into the unique properties of Kecai yaks.

Genome-wide analyses revealed that over 50% of the selected regions of the Kecai and Gannan yak genomes were under positive selection, while further supporting the classification of Kecai yaks as a distinct yak subgroup. GO analyses of genes annotated in these selected regions revealed their enrichment in biological processes and molecular functions related to transfer between glycolipid membranes, including glycolipid transfer protein (GLTP) involved in ceramide transport, membrane interlipid transfer, intermembrane lipid transfer activity, ceramide 1-phosphate binding, and ceramide 1-phosphate transporter activities. GLTP is a soluble 23.8 kDa protein that can accelerate selective intermembrane glycolipid transfer [28,29,30]. Glycosphingolipids (GSLs) function to stimulate DNA synthesis, cell growth, differentiation, and the sorting and transport of proteins, in addition to influencing eukaryotic processes, such as adhesion, cell–cell recognition, and development [31,32,33]. GLTP can bind and transfer a wide range of GSLs, functioning as a de facto sensor for levels of GlcCer within cells and regulating normal cell sphingolipid homeostasis. Selected genes identified in this study were significantly enriched in the regulation of adipose tissue development [34,35]. The metabolism of both fat and muscle tissue is critical for the maintenance of an appropriate energy balance in yaks. Given their prolonged evolution under high-altitude alpine conditions, yaks exhibit a unique body structure and corresponding genetic changes that govern adipose tissue development [36]. The genetic differences identified in this study suggest that Kecai yaks exhibit greater advantages with respect to their ability to adapt to the environment of alpine regions relative to other domesticated yaks.

KEGG enrichment analyses indicated that genes in selected regions of the genome were most strongly enriched in the Staphylococcus aureus infection, estrogen signaling, and glycosaminoglycan biosynthesis–chondroitin sulfate/pidan sulfate pathways. Amino acids are essential taste-active compounds that play a central role in establishing the flavor of meat. The unsaturated fatty acid biosynthesis, linolenic acid metabolism, arachidonic acid metabolism, and linoleic acid metabolism pathways detected in this study are involved in muscle fat development, and ACOX1 is an acyl-CoA oxidate associated with fatty acid oxidative decomposition. ACOX1 is a highly conserved and unique enzyme that is highly expressed in the liver, with lower levels of expression being evident in the kidney, brain, and adipose tissue consistent with its importance in the context of fat metabolism [37,38]. Many studies have demonstrated a close relationship between the ACOX1 genes and loci related to quantitative traits that influence daily gain, birth weight, backfat thickness, and fatty acid composition [39,40,41]. As such, ACOX1 represents a candidate gene related to fat deposition that plays important roles in fatty acid metabolism and may be associated with the excellent muscle fat deposition and meat quality observed in Kecai yaks.

## 5. Conclusions

In summary, a whole-genome resequencing strategy was used in this paper to explore the genetic diversity and population structure of Kecai yaks. The SNP markers used for these analyses revealed that Kecai yaks are genetically distinguishable from wild and Gannan yak populations. These Kecai yaks also exhibited evidence of the functional enrichment of positively selected genes relative to Ganna yaks, suggesting that these genes may play a role in the meat quality of these animals and their adaptation to alpine environments. The comprehensive analysis showed that the Kecai yak has unique genetic characteristics. This study provides a theoretical basis for the protection and utilization of Kecai yak resources.

## Figures and Tables

**Figure 1 animals-12-02682-f001:**
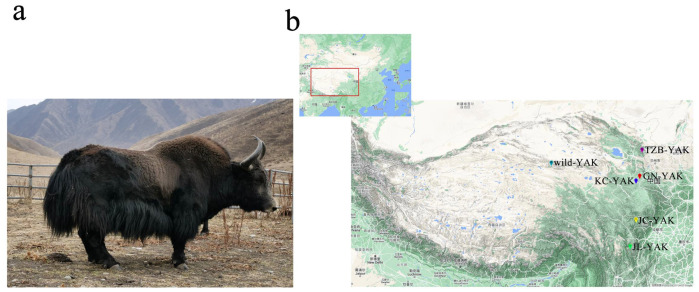
The geographical distribution of yaks selected in this study. (**a**) Keicai yak. (**b**) Sampling distribution map. GN: Gannan yak; KC: Kecai yak; TZB: Tianzhu white yak; JC: Jinchuan yak; JL: Jiulong yak.

**Figure 2 animals-12-02682-f002:**
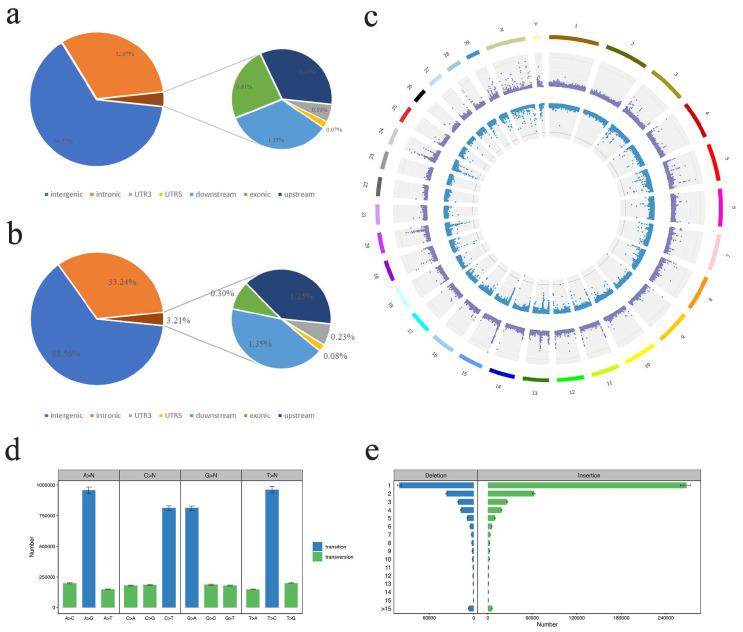
Kecai yak SNP and InDel statistics. (**a**) SNP annotation results. (**b**) InDel annotation results. (**c**) SNP and InDel distributions on individual chromosomes, where the first circle represents the yak chromosome, the purple dots in the second circle represent the SNP density, and the blue dots in the third circle represent the indel density. (**d**) SNP transition and transversion distributions. (**e**) InDel insertion and deletion statistics.

**Figure 3 animals-12-02682-f003:**
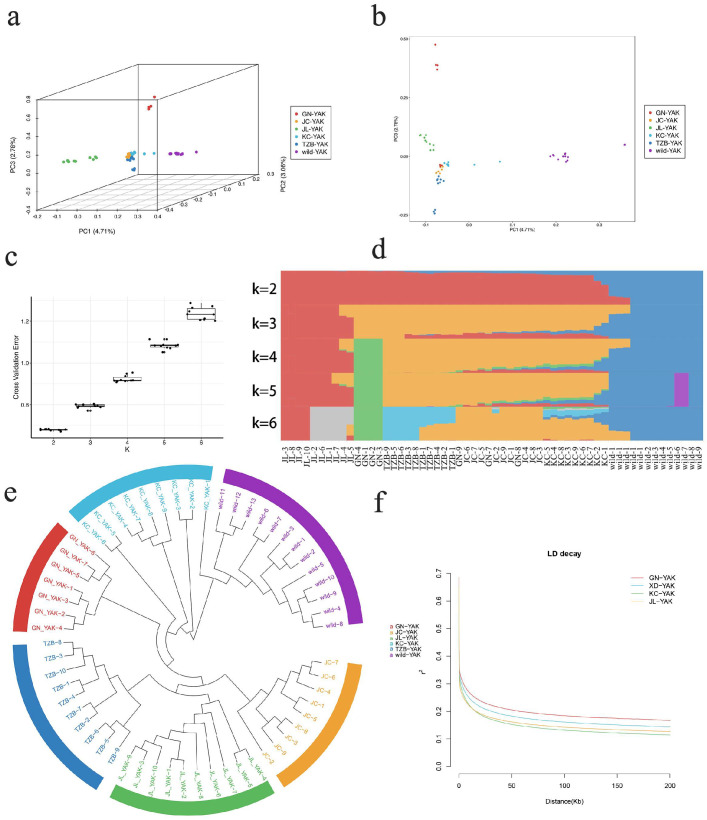
Yak population stratification analyses. (**a,b**) Yak principal component analysis results represented in three dimensions (**a**) and two dimensions (**b**). Different colors correspond to different yak varieties. (**c**) Cross−validation error results at different k values. (**d**) Yak population structure analysis. (**e**) Yak phylogenetic analysis. (**f**) Yak LD decay analyses.

**Figure 4 animals-12-02682-f004:**
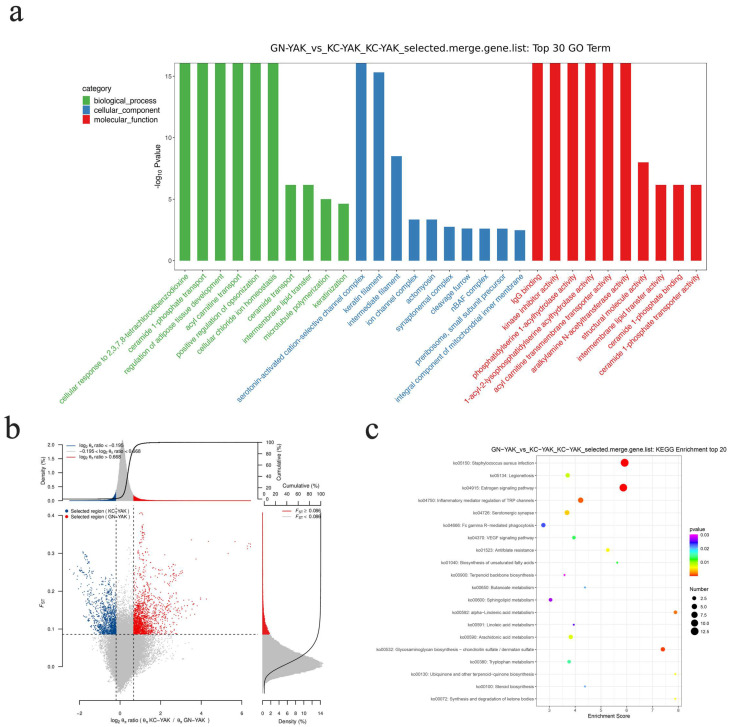
Analyses of the enrichment and functional annotation of the Kecai yak genome under genetic selection. (**a**) GO enrichment results for selected genes detected in the Kecai yak genome. (**b**) A location map comparing selected areas of the genome between Kecai and Gannan yaks. The selected area was determined based on the top 5% area of Fst values and the θπ ratio, with a window of 200 kb and a step size of 20 kb. Blue data points correspond to the selected area identified in Kecai yak samples. Red data points correspond to the selected area identified in Gannan yak samples. (**c**) KEGG pathway enrichment results for the genes in the selected regions of the Kecai yak genome.

## Data Availability

The genome data of wild yak were selected from the breeding genome data published by the European Nucleotide Archive (EMBL-EBI) (Accession number: PRJNA285834), and the sequencing data of Kecai yak were uploaded to the Sequence Read Archive (SRA) (Accession number: PRJNA842787).

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
