# Peer review of "Whole-Genome Resequencing Highlights the Unique Characteristics of Kecai Yaks"

_animals, 2022, doi:10.3390/ani12192682_

Round 1

Reviewer 1 Report

The authors have presented a comprehensive review of published studies with Yaks in the introduction. The authors should improve the quality of the images and increase the text size in the figures. The results obtained by the authors and their statements sometimes seem contradictory. The authors can improve the discussion section.

Find my considerations below.

Line 25: to their high fecundity and the flavorful meat that they produce -> to their high fecundity and the flavorful meat

Lines 27, 45: these yaks -> Kecai yaks

In abstract: verify when to replace genes for regions.

Line 48:  Exclude ‘to date’.

Line 70: GRIN2D in italic.

Line 75: Verify ‘nutrient-rich mean’.

Line 79: These Kecai yaks à The Kecai yaks.  Correct ‘large exhibit high fecundity’.

Line 83: Correct ‘have yet to be defined’.

Figure 1. text within the image is hard to read.

Line 108: collected yak blood samples -> individual blood sample

Item 2.2. Replace ‘Those samples’, ‘these samples’, ‘These data’. Avoid to use those, this, these in the text.

Item 2.7. Include citations for the statistics (θπ ratio and FST). There are many ways to compute Fst, which one was used? Is the θπ ratio related to the nucleotide diversity?

Line 182: kB -> kb

Lines 184 and 188: What do the authors mean with ‘therein’? Rephrase the sentence.

Lines 193-195: Check text in ‘Overall, 95.23% of reads with a quality value ≥20 bases (sequencing error rate < 0.01), with a GC content of 43.14% consistent with high-quality sequencing analyses.’

Line 195: ‘98.75%%’

Line 208: Change the text in ‘Of these InDels’.

Figure 2: The text is too small. Provide more details in the figure legend for item c. The gray text in the blue part of the image is difficult to read.

Line 223: Typo in ‘tak populations’.

Line 232: Specify ‘these yaks’.

Line 234: Correct ‘When n k ≥ 4’.

Lines 239: ‘These results suggest that relative to these other 239 yak populations, Kecai yaks have been subject to a lesser degree of domestication and to 240 less intense selection (Fig. 3f).’ Why?

Figure 3: Not sure if there was a meaning for the abbreviations (GN, JC, JL, KC, etc) anywhere in the text. I recommend to provide a legend for the abbreviations in the figure’s subtitle. It is impossible to read the text in items d and e. What explains the genetic differentiation between two subgroups of Gannan yaks showed in item b? It seems Kecai breed is closely related to one subpopulation of Gannan opposing the authors statements in item 3.3. How do the authors explain that?

Line 251: this FST value -> the average Fst value

Line 264: what do the authors mean with ‘consistent with classification as a separate breed’?

Line 279: Gene name should appear in italic (ACOX1).

It is impossible to read items b and c in figure 4. Can the authors improve the quality of the images or increase the text size?

Line 282: ‘of areas’? What do the authors mean? Rewrite the sentence.

Line 283: region genes -> genes detected

Figure 4: Include the red dots in the legend for item b.

Line 308: ‘Some Kecai yaks were genetically similar to wild yaks and completely separated from Gannan yaks, indicating that they are distinct from other domesticated yak populations.’ This statement is controversial from what is shown in the PCA plot. Can the authors review it? If Kecai was originated from Gannan and wild yaks, why was Kecai closer related to other breeds? From where the shared ‘genetic background’ with other breeds comes from? I suggest to explain it in the text. ‘PCA analyses revealed clustering between Kecai, Jiulong, Jinchuan, Tianzhu white, and wild yak samples with thiss cluster being separated from the cluster comprised of Gannan yak samples.’

Line 315: typo. ‘Kecia’

Line 365: ‘Kecai yaks were genetically distinguishable from wild and Gannan yak populations’ versus ‘Kecai yaks were generated through the cross-breeding of wild and Gannan yaks.’ (in line 311).

Conclusions: The quality of the text and contents in conclusion can be improved.

Author Response

  1. Line 25 = to their high fecundity and the flavorful meat that they produce -> to their high fecundity and the flavorful meat.

Response 1: Thank you for your comments. We have replaced “to their high fecundity and the flavorful meat that they produce” with “to their high fecundity and the flavorful meat” in the text in line 25.

  1. Line 27, 45 = these yaks -> Kecai yaks

Response 2: Thank you for your comments. We have replaced “these yaks” with “Kecai yaks” in the text in line 27, 44.

  1. In abstract: verify when to replace genes for regions.

Response 3: Thank you for your comments. The article has been revised as requested. We have replaced “Kecai yak genes that had been positively selected relative to those of closely related Gannan yaks were also annotated, with functional enrichment analyses of these genes suggesting that such selection in Kecai yaks was associated with the adaptation to alpine environments and the deposition of muscle fat.” with “Anno-tation and functional enrichment analysis of genes related to positive selection in Kecai yak, the results show that such selection in Kecai yaks was associated with the adaptation to alpine envi-ronments and the deposition of muscle fat.” in the text in line 32-35.

  1. Line 48 = Exclude ‘to date’.

Response 4: Thank you for your comments. We have removed “to date” in line 47.

  1. Line 70 = GRIN2D in italic.

Response 5: Thank you for your comments. We have replaced “GRIN2D” with “GRIN2D” in the text in line69.

  1. Line 75 = Verify ‘nutrient-rich mean’.

Response 6: Thank you for your comments. The article has been revised as requested in line 75.

  1. Line 79 = These Kecai yaks -> The Kecai yaks. Correct ‘large exhibit high fecundity’.

Response 7: Thank you for your comments. We have replaced “These Kecai yaks” with “The Kecai yaks” in the text in line 79.

  1. Line 83 = Correct ‘have yet to be defined’.

Response 8: Thank you for your comments. The article has been revised as requested in line 82.

  1. Figure 1. text within the image is hard to read.

Response 9: Thank you for your comments. Figure has been modified as requested.

  1. Line 108 = collected yak blood samples -> individual blood sample

Response 10: Thank you for your comments. We have replaced “collected yak blood samples” with “individual blood sample” in the text in line 108.

  1. Item 2.2 = Replace ‘Those samples’, ‘these samples’, ‘These data’. Avoid to use those, this, these in the text.

Response 11: Thank you for your comments. The article has been revised as requested.

  1. Item 2.7 = Include citations for the statistics (θπ ratio and FST). There are many ways to compute Fst, which one was used? Is the θπ ratio related to the nucleotide diversity?

Response 12: Thank you for your comments. We have added supporting references. In this study, the θπ ratio and Fst results were obtained by VCFtools and sliding window calculation, with a window of 200kb and a step size of 20kb. Genetic diversity (θπ value), which is used to measure nucleotide diversity within a population, can also be used to deduce evolutionary relationships.

  1. Line 182 = kB -> kb

Response 13: Thank you for your comments. We have replaced “kB” with “kb” in the text in line 184.

  1. Line 184, 188 = What do the authors mean with ‘therein’? Rephrase the sentence.

Response 14: Thank you for your comments. We have replaced “Those regions in the top and bottom 5% of ln θπ ratio values and the top 5% of FST values were identified as selected regions of the genome, and the target genes therein were subject to functional enrichment analyses using Gene Ontology (GO) and Kyoto Encyclopedia of Genes and Genomes (KEGG) functional annotation tools.” with “Those regions in the top and bottom 5% of log2 θπ ratio values and the top 5% of Fst values were identified as selected regions of the genome, and the target genes were subject to functional enrichment analyses using Gene Ontology (GO) and Kyoto Ency-clopedia of Genes and Genomes (KEGG) functional annotation tools.” in the text.

  1. Line 193-195 = Check text in ‘Overall, 95.23% of reads with a quality value ≥20 bases (sequencing error rate < 0.01), with a GC content of 43.14% consistent with high-quality sequencing analyses.’

Response 15: Thank you for your comments. We have replaced “Overall, 95.23% of reads with a quality value ≥20 bases (sequencing error rate < 0.01), with a GC content of 43.14% consistent with high-quality sequencing analyses” with “Overall, 95.23% of reads with a quality value ≥20 bases (sequencing error rate < 0.01), with a GC content of 43.14%, with good sequencing quality” in the text.

  1. Line 195 = ‘75%%’

Response 16: Thank you for your comments. We have replaced “98.75%%” with “98.75%” in the text in line 195.

  1. Line 208 = Change the text in ‘Of these InDels’.

Response 17: Thank you for your comments. We have replaced “Of these InDels, 892,747 (63.35%) were located in intergenic regions, while 466,874 (33.13%) were located in intronic regions. Overall, just 4,178 (0.30%) InDels were located in exonic regions (Fig. 2b), of which 1,217 induced frameshift deletions, 2,108 induced frameshift insertions, and 432 and 342 respectively induced non-frameshift deletions and non-frameshift insertions (Supplementary Table S4).” with “Of these InDels, 63.35% of the indels are located in the intergenic region, 33.13% in the intron region, and only 0.30% in the exon region (Fig. 2b). Among the indels located in the exon region, 1,217 induced frameshift deletions, 2,108 induced frameshift inser-tions, and 432 and 342 respectively induced non-frameshift deletions and non-frameshift insertions (Supplementary Table S4).” in the text in line 210-214.

  1. Figure 2: The text is too small. Provide more details in the figure legend for item c. The gray text in the blue part of the image is difficult to read.

Response 18: Thank you for your comments. Figure has been revised as requested.

  1. Line 223 = Typo in ‘tak populations’

Response 19: Thank you for your comments. We have replaced “tak populations” with “yak populations” in the text in line 227.

  1. Line 232 = Specify ‘these yaks’

Response 20: Thank you for your comments. We have replaced “In further analyses of population structure, at k=2 these yaks were separated into two groups” with “In further analyses of population structure, at k=2 all of yaks were separated into two groups” in the text in line 236.

  1. Line 232 = Correct ‘When n k ≥ 4’.

Response 21: Thank you for your comments. We have replaced “When n k ≥ 4” with “When k ≥ 4” in the text in line 238.

  1. Line 239 =‘These results suggest that relative to these other 239 yak populations, Kecai yaks have been subject to a lesser degree of domestication and to 240 less intense selection (Fig. 3f). ’ Why?

Response 22: Thank you for your comments. Linkage disequilibrium (LD): refers to the phenomenon that in a population, the frequency of simultaneous inheritance of two genes at different loci is significantly higher than the expected random frequency; that is, when two genes located on the same chromosome ( A, B) When the probability of co-existence is greater than the probability of co-occurrence due to random distribution in the population, the two genes are said to be in linkage disequilibrium (LD) state, and the r2 (LD coefficient) value is usually used to measure linkage disequilibrium degree, its value fluctuates from 0 to 1. r2=0 means that the two loci are completely unrelated, and the distribution of the haplotypes of these two loci in the population is random (observed value = expected value). r2 = 1 means that the two loci are completely related, and a certain genotype (A) only co-occurs with a specific genotype (B). In general, the closer two loci are on the genome, the stronger the correlation and the larger the LD coefficient. Conversely, the smaller the LD coefficient is. This law is usually presented using an LD decay graph, which uses a graph to present the process of decreasing the average LD coefficient between molecular markers on the genome as the distance between markers increases.

In linkage disequilibrium analysis, the LD value of the wild species is usually lower, while the LD value of the domesticated species will be larger due to the effect of positive selection, which increases the LD degree of the selected region on the genome. Factors such as natural selection and genetic drift will lead to a decrease in population genetic diversity and a slower rate of decay. In addition, selected regions of the genome decay at a slower rate than normal regions.

  1. Figure 3: Not sure if there was a meaning for the abbreviations (GN, JC, JL, KC, etc) anywhere in the text. I recommend to provide a legend for the abbreviations in the figure’s subtitle. It is impossible to read the text in items d and e. What explains the genetic differentiation between two subgroups of Gannan yaks showed in item b?  It seems Kecai breed is closely related to one subpopulation of Gannan opposing the authors statements in item 3.3. How do the authors explain that?

Response 23: Thank you for your comments. The article has been revised as requested. The Gannan yak may be due to the convergence of yaks from different refuges after the end of the last ice peak. The experimental sampling samples may involve the descendants of yaks gathered from the two refuges, so there is a certain genetic differentiation between the two groups. The distribution area of Gannan yak is wide, and the distribution area of Kecai yak is small. It may be evolved from a branch of Gannan yak, so it is closely related to a subgroup of Gannan yak.

  1. Line 251 = this FST value -> the average Fst value

Response 24: Thank you for your comments. We have replaced “this FST value” with “the average Fst value” in the text in line 255.

  1. Line 264 =what do the authors mean with ‘consistent with classification as a separate breed’?

Response 25: Thank you for your comments. We have replaced “with values above these thresholds being indicative of the influence of positive selection consistent with classification as a separate breed.” with “above these thresholds indicate positive selection and can be divided into groups with unique characteristics.” in the text in line 266-268.

  1. Line 279 = Gene name should appear in italic (ACOX1).

Response 26: Thank you for your comments. We have replaced “ACOX1” with “ACOX1” in the text in line 282.

  1. It is impossible to read items b and c in figure 4. Can the authors improve the quality of the images or increase the text size?

Response 27: Thank you for your comments. Figure has been revised as requested.

  1. Line 282 = ‘of areas’? What do the authors mean? Rewrite the sentence.

Response 28: Thank you for your comments. We have replaced “Analyses of the enrichment and functional annotation of areas of the Kecai yak genome under genetic selection” with “Analyses of the enrichment and functional annotation of the Kecai yak genome under genetic selection” in the text in line 285.

  1. Line 283 = region genes -> genes detected

Response 29: Thank you for your comments. We have replaced “region genes” with “genes detected” in the text in line 286.

  1. Figure 4: Include the red dots in the legend for item b.

Response 30: Thank you for your comments. The article has been revised as requested. Red data points correspond to the selected area identified in Gannan yak samples.

  1. Line 308 = ‘Some Kecai yaks were genetically similar to wild yaks and completely separated from Gannan yaks, indicating that they are distinct from other domesticated yak populations’ This statement is controversial from what is shown in the PCA plot. Can the authors review it? If Kecai was originated from Gannan and wild yaks, why was Kecai closer related to other breeds? From where the shared ‘genetic background’ with other breeds comes from?  I suggest to explain it in the text. ‘PCA analyses revealed clustering between Kecai, Jiulong, Jinchuan, Tianzhu white, and wild yak samples with thiss cluster being separated from the cluster comprised of Gannan yak samples.’

Response 31: Thank you for your comments. The article has been revised as requested. Kecai yak is closely related to some Gannan yak and some other domestic yak. At the same time, Fig. 3b shows that Kecai yak is between some Gannan yak and wild yak, which is consistent with the origin of Kecai yak from Gannan yak and wild yak.

  1. Line 315 = typo. ‘kecia’

Response 32: Thank you for your comments. We have replaced “kecia” with “kecai” in the text.

  1. Line 365 = ‘Kecai yaks were genetically distinguishable from wild and Gannan yak populations’ versus ‘Kecai yaks were generated through the cross-breeding of wild and Gannan yaks’ (in line 311).

Response 33: Thank you for your comments. We have replaced “Kecai yaks were generated through the cross-breeding of wild and Gannan yaks” with “Kecai yaks were genetically distinguishable from wild and Gannan yak populations” in the text in line 313.

Reviewer 2 Report

Kang et al. offered an insight into the genetic characteristics of Kecai yaks. The manuscript is well written and all selected methods are appropriate. The authors also provided a deep discussion about the results. The manuscript is almost ready for publication.

Line 43-47: Please add some supporting references.

Line 144: Which reference genome did the authors use?

Line 146-154: Some abbreviations should be defined even if they are commonly used in the variant calling/filtering steps.

Line 178: Why did the authors choose θπ ratio and Fst methods instead of haplotype-based methods? Might add a short justification of method choice.

Line 181: is it ln θπ ratio or log10 θπ ratios? And the ratio of what?

Provide which tool is used for geneset enrichment.

Figure 2: the legend for the figure is not visible, please increase the resolution or change the font size. Similar comments for Figures 3 and 4, the authors could also increase the figure size

Author Response

  1. Line 43-47 = Please add some supporting references.

Response 1: Thank you for your comments. We have added supporting references in line 46.

Guo, S.; Savolainen, P.; Su, J.; Zhang, Q.; Qi, D.; Zhou, J.; Zhong, Y.; Zhao, X.; Liu, J. Origin of Mitochondrial DNA Diversity of Domestic Yaks. BMC evolutionary biology 2006, 6, 73–73, doi:10.1186/1471-2148-6-73.

Wang, Z.; Shen, X.; Liu, B.; Su, J.; Yonezawa, T.; Yu, Y.; Guo, S.; Ho, S.Y.W.; Vilà, C.; Hasegawa, M.; et al. Phylogeographical Analyses of Domestic and Wild Yaks Based on Mitochondrial DNA: New Data and Reappraisal: Phylogeographical Patterns of Yaks. Journal of biogeography 2010, 37, 2332–2344, doi:10.1111/j.1365-2699.2010.02379.x.

  1. Line 144 = Which reference genome did the authors use?

Response 2: Thank you for your advice. The reference genome we used was LU_Bosgru_v3.0. The article has been revised as requested in line 145.

  1. Line 146-154 = Some abbreviations should be defined even if they are commonly used in the variant calling/filtering steps.

Response 3: Thank you for your advice. We have replaced “(1) GATK filtering parameters for loci exclusion: QD < 2.0 || MQ < 40.0 || FS > 60.0 || SOR > 3.0 || MQRankSum < -12.5 || ReadPosRankSum < -8.0;” with “(1) The Genome Analysis Toolkit (GATK) filtering parameters for loci exclusion: QualByDepth (QD) < 2.0 || RMSMappingQuality (MQ) < 40.0 || FisherStrand (FS) > 60.0 || StrandOddsRatio (SOR) > 3.0 || MappingQualityRankSumTest (MQRankSum) < -12.5 || ReadPosRankSum < -8.0;” in the text.

  1. Line 178 = Why did the authors choose θπ ratio and Fst methods instead of haplotype-based methods? Might add a short justification of method choice.

Response 4: Thank you for your advice. The θπ ratio and Fst method was chosen because it is the most common selective removal analysis method.

  1. Line 181 = is it ln θπ ratio or log10 θπ ratios?

Response 5: Thank you for your advice. We have replaced “The ln θπ ratio and FST values were calculated using VCFtools [21], with a window size of 100 kB and a step size of 10 kB. Those regions in the top and bottom 5% of ln θπ ratio values and the top 5% of FST values were identified as selected regions of the genome, and the target genes therein were subject to functional enrichment analyses using Gene Ontology (GO) and Kyoto Encyclopedia of Genes and Genomes (KEGG) functional annotation tools” with “The log2 θπ ratio and Fst values were calculated using VCFtools [25], with a window size of 100 kb and a step size of 10 kb. Those regions in the top and bottom 5% of log2 θπ ratio values and the top 5% of Fst values were identified as selected regions of the genome, and the target genes were subject to functional enrichment analyses using Gene Ontology (GO) and Kyoto Encyclopedia of Genes and Genomes (KEGG) func-tional annotation tools.” in the text.

  1. Provide which tool is used for geneset enrichment.

Response 6: Thank you for your advice. Gene enrichment analysis is done by writing R script by myself

  1. Figure 2: the legend for the figure is not visible, please increase the resolution or change the font size. Similar comments for Figures 3 and 4, the authors could also increase the figure size.

Response 7: Thank you for your advice. Figure has been revised as requested.

Round 2

Reviewer 1 Report

Line 32: an incomplete sentence was added. ‘Annotation and functional enrichment analysis of genes related to positive selection in Kecai yak’

The conclusion section seems short and does not summarize the main information in the manuscript. I would suggest improvements.

Author Response

  1. Line 32: an incomplete sentence was added. ‘Annotation and functional enrichment analysis of genes related to positive selection in Kecai yak’

Response 1: Thank you for your comments. We have replaced “Annotation and functional enrichment analysis of genes related to positive selection in Kecai yak” with “In this study, marker and functional enrichment analysis of genes related to positive selection in Kecai yak was carried out” in the text in line 32.

  1. The conclusion section seems short and does not summarize the main information in the manuscript.

Response 2: Thank you for your comments. We have replaced “In summary, a whole genome resequencing strategy was herein used to explore the genetic diversity and population structure of Kecai yaks. The SNP markers used for these analyses revealed that Kecai yaks were genetically distinguishable from wild and Gannan yak populations. The comprehensive analysis showed that the Kecai yak has unique genetic characteristics.” with “In summary, a whole genome resequencing strategy was herein used to explore the genetic diversity and population structure of Kecai yaks. The SNP markers used for these analyses revealed that Kecai yaks were genetically distinguishable from wild and Gannan yak populations. These Kecai yaks also exhibited evidence of the functional enrichment of positively selected genes relative to Ganna yaks, suggesting that these genes may play a role in the meat quality of these animals and their adaptation to alpine environments. The comprehensive analysis showed that the Kecai yak has unique genetic characteristics. This study provides a theoretical basis for the protection and utilization of Kecai yak resources.” in the text in line 360-368.
